# Biomarkers in Alzheimer’s Disease: Are Olfactory Neuronal Precursors Useful for *Antemortem* Biomarker Research?

**DOI:** 10.3390/brainsci14010046

**Published:** 2024-01-02

**Authors:** Valeria Santillán-Morales, Norberto Rodriguez-Espinosa, Jesús Muñoz-Estrada, Salvador Alarcón-Elizalde, Ángel Acebes, Gloria Benítez-King

**Affiliations:** 1Laboratory of Neuropharmacology, Clinical Research, Instituto Nacional de Psiquiatría Ramón de la Fuente Muñiz, Mexico City 14370, Mexico; santillan.val.29@gmail.com (V.S.-M.); salalel@imp.edu.mx (S.A.-E.); 2Department of Neurology, University Hospital Nuestra Señora de Candelaria, 38010 Tenerife, Spain; nrodesp@ull.edu.es; 3Department of Internal Medicine, Dermatology and Psychiatry, Faculty of Health Sciences, University of La Laguna (ULL), 38200 Tenerife, Spain; 4Department of Computational Biomedicine, Cedars Sinai Medical Center, Los Angeles, CA 90069, USA; josedejesus.munozestrada@cshs.org; 5Department of Basic Medical Sciences, Institute of Biomedical Technologies (ITB), University of La Laguna (ULL), 38200 Tenerife, Spain

**Keywords:** olfactory neuronal precursors, Alzheimer’s disease, biomarkers, olfactory neuroepithelium

## Abstract

Alzheimer’s disease (AD), as the main cause of dementia, affects millions of people around the world, whose diagnosis is based mainly on clinical criteria. Unfortunately, the diagnosis is obtained very late, when the neurodegenerative damage is significant for most patients. Therefore, the exhaustive study of biomarkers is indispensable for diagnostic, prognostic, and even follow-up support. AD is a multifactorial disease, and knowing its underlying pathological mechanisms is crucial to propose new and valuable biomarkers. In this review, we summarize some of the main biomarkers described in AD, which have been evaluated mainly by imaging studies in cerebrospinal fluid and blood samples. Furthermore, we describe and propose neuronal precursors derived from the olfactory neuroepithelium as a potential resource to evaluate some of the widely known biomarkers of AD and to gear toward searching for new biomarkers. These neuronal lineage cells, which can be obtained directly from patients through a non-invasive and outpatient procedure, display several characteristics that validate them as a surrogate model to study the central nervous system, allowing the analysis of AD pathophysiological processes. Moreover, the ease of obtaining and harvesting endows them as an accessible and powerful resource to evaluate biomarkers in clinical practice.

## 1. Introduction

Alzheimer’s disease (AD) is a progressive neurodegenerative disease that represents a global health problem. AD is the leading cause of dementia, representing about 70% of cases. In the world, there are 50 million people with dementia, and it is estimated that this figure will triple in the next two decades [1]. AD mainly affects older adults, generating a slow and progressive deterioration of cognitive ability and interfering with normal brain functions. Due to the fact that neural damage initially affects areas of the brain involved in memory, language, and thinking, early symptoms of AD can be manifested as difficulty in remembering recent conversations, names, or events. Depression and apathy are also early manifestations. Subsequently, problems begin in communication, judgment, and confusion. Eventually, brain damage covers areas involved in basic functions such as walking, swallowing, and speaking [2]. The neurodegenerative process begins years before the appearance of the first symptoms. In clinical settings, diagnosis is probable or possible, and it is made by clinical criteria, mainly those of the National Institute on Aging and Alzheimer’s Associations (NIA-AA) and the Diagnostic and Statistical Manual of Mental Disorders (DSM-5) criteria [3,4]. Frequently, the symptoms are subtle and are usually ignored, diminished, and undiagnosed, which delays proper treatment and care [2]. At the stage in which it is typically diagnosed, there has already been significant neuronal loss in many brain areas. Therefore, the detection of biological markers is necessary to increase the opportunity for early diagnosis of AD.

In this review, we will address the importance of studying biomarkers in AD as a tool that could improve the way we approach this condition. In order to study and propose biomarkers, it is necessary to know the pathophysiology of the disease; therefore, some of the processes that contribute to AD and their role as biomarkers will be summarized here. Likewise, we also propose to employ olfactory neuronal precursors (ONPs) derived from olfactory neuroepithelium (ONE) as a new and valuable model for the study of AD biomarkers. The ONE cultured ex vivo contains cells of neuronal lineage obtained from patients and healthy individuals, which allows the study of the cellular and molecular mechanisms involved in neurodegenerative diseases such as AD.

## 2. Current Biomarkers in Alzheimer’s Disease

### 2.1. The Need for Alzheimer’s Disease Biomarkers

The National Institutes of Health (NIH) defines a biomarker as a characteristic that is objectively measured and evaluated as an indication of normal biological processes, pathogenic processes, or pharmacologic responses to a therapeutic intervention [5]. Biomarkers are useful tools to screen, characterize, rule out, diagnose, stage, and monitor diseases, as well as to inform prognosis, individualize therapeutic interventions, predict adverse drug reactions, and identify cell types [6]. One of the most important challenges of neurodegenerative diseases such as AD is the subjective approaches to perform the diagnosis that delays the opportunity for adequate and timely management and treatment. Thus, the detection of biomarkers at the early stages of the disease is indispensable to increase the opportunity to delay AD progression or even mild cognitive impairment (MCI), which are markers of early detection.

The NIA-AA criteria for AD are classified into three categories: (1) probable AD dementia, (2) possible AD dementia, and (3) probable or possible AD dementia with evidence of biomarkers for the pathophysiological process of AD. The first two are established in clinical settings, and the last one is established for research purposes [4]. Biomarkers for the pathophysiological process of AD included in these criteria are subdivided into two main classes: markers of amyloid deposition and markers of neuronal damage. The former includes a decrease in cerebrospinal fluid (CSF) amyloid beta levels and positive positron emission tomography (PET) amyloid imaging. Increased CSF tau (including total tau and phosphorylated tau) decreased fluorodeoxyglucose positron emission tomography (FDG-PET) consumption in the temporoparietal cortex, and disproportionate atrophy in the medial, basal, and lateral temporal lobe temporoparietal cortex and in the medial parietal cortex are markers of neuronal damage. Although the presence of these biomarkers increases the certainty of the diagnosis of AD, the use of these biomarkers as a routine diagnostic test is not recommended [4]. One of the reasons is the limited access to these biomarkers in common clinical settings. Table 1 summarizes the biomarkers recognized in the diagnostic criteria of the NIA-AA and DSM-5, as well as their use and interpretation in clinical practice.

### 2.2. Brain Imaging Markers

Structural imaging studies in AD show atrophy in many brain areas. This atrophy begins in the medial temporal lobe, in areas including the entorhinal cortex and hippocampus. Structures of the limbic system, such as the olfactory bulb tract, thalamus, and amygdala, are also affected. Subsequently, there is a loss of the gray matter of the parahippocampus and medial temporal gyrus. The frontal, parietal, and temporal areas of the brain also suffer volume reduction [7]. These structural changes can be evaluated by magnetic resonance imaging (MRI). MRI is a non-invasive imaging procedure that does not require the injection of a contrast agent or radiation exposure. However, the analysis and rating of these images are qualitative and subjective, causing different results between evaluators. Although there are several MRI techniques, only traditional structural imaging is currently recommended for routine use in clinical settings. Automated methods that measure hippocampal volume have been proposed to decrease the variability of results, achieving the correct classification rate for AD and cases without cognitive impairment in about 80% and 65% for stable and progressive MCI cases [8]. 

PET imaging studies use radiotracers that bind to insoluble fibrillar forms of Aβ40 and Aβ42 to visualize amyloid deposition in the brain in vivo [9]. PET amyloid detection is highest in the frontal cortex, followed by the parietal, temporal, and occipital cortex, and striatum (Table 2) [10]. The amyloid regional distribution detected with PET correlates to the distribution of amyloid deposits observed in autopsies [11,12,13].

FDG-PET detects the metabolism of glucose in patients’ brains. In patients with early AD, the areas of glucose hypometabolism have been commonly observed in the parietotemporal association cortices, posterior cingulate cortex, and the precuneus [9]. In a meta-analysis that included 119 studies, FDG-PET was found to have a sensitivity of 90% and a specificity of 89% for identifying AD when compared to controls without dementia. In addition, FDG-PET was able to discriminate AD dementia from other causes of dementia with a sensitivity of 92% and a specificity of 78% [30].

### 2.3. Postmortem Markers

A diagnosis of definite Alzheimer’s disease requires histopathologic confirmation obtained from a biopsy or autopsy: the presence of neuritic plaques composed of Aβ aggregates and neurofibrillary tangles (NFTs) formed from hyperphosphorylated tau protein (Table 2) [15,31].

### 2.4. Markers in Biological Fluids (Cerebrospinal Fluid and Plasma)

The CSF is in direct contact with the CNS and is considered to reflect the pathological processes that occur in the brain. For these reasons, CSF has been studied as a resource to study biomarkers in AD. The main proposed biomarkers are amyloid beta peptide (specifically Aβ42), total tau (t-tau), and phosphorylated tau (p-tau181 and p-tau231) [32].

Various studies have shown that whereas Aβ levels are decreased in the CSF of patients with AD, tau (t-tau and p-tau) levels are, in contrast, increased (Table 2) [14]. CSF biomarkers are strongly associated with AD diagnosis and cortical Aβ deposition [33]. Correlations exist between CSF p-tau concentrations and scores of NFTs and hyperphosphorylated-tau load in all neocortical regions [34]. The reduction in Aβ42 is about 50% in AD patients compared to age-matched controls without dementia [35]. The value of Aβ42 in the CSF was analyzed in a systematic review where four studies were included, and a median specificity of 64% and sensitivity of 81% were reported [36]. Total tau CSF levels are about three times higher in AD patients than in controls [32]. Accuracy increases when both CSF biomarkers are combined, as shown by a study with neuropathological confirmation of the diagnosis, which showed that the p-tau/Aβ42 ratio has a sensitivity of 91.6% and a specificity of 85.7% for AD diagnosis [32].

In clinical settings, biomarkers are particularly relevant for both the diagnosis and follow-up of patients with suspected AD. In the very near future, they will also be decisive as markers of response and prognosis in the face of new treatments based on immunotherapy against amyloid peptides [37,38]. Biomarkers for diagnostic support are essential to be directly related to key aspects of the AD pathological process, i.e., abnormal processing of amyloid beta peptide and total and phosphorylated tau protein. As mentioned above, CSF determination of Aβ peptide fractions 1–40 and 1–42, together with total tau and phosphorylated tau proteins, currently offer high diagnostic accuracy, both in terms of sensitivity, specificity, and predictive values.

Recently, diagnostic biomarkers in plasma have also been developed as a method to facilitate and reduce the costs of population screening. The performance of the Aβ1-42/Aβ1-40 index and 181p-tau together with plasma glial fibrillary acidic protein (GFAP) has generated great interest [39]. Compared to necropsy as the gold standard, PET and CSF-related biomarkers in combination showed high concordance with the pathological stage; alone, they showed high accuracy in discriminating autopsy-confirmed AD. However, the plasma-related biomarkers alone showed better discriminative performance only when combined with the apolipoprotein E (APO) Eε4 genotype [40]. In this context, biomarkers that accurately reflect disease progression and have prognostic capabilities are also needed. In AD, the cognitive reserve accumulated throughout life, based on the level of education, professional qualifications, hobbies, and the intensity of social relationships, cushions the severity of the clinical manifestations, and it is not uncommon for individuals who, despite bearing a high lesional burden typical of the advanced stages of the disease, show few symptoms with little impact on their daily lives [41]. Due to the amyloid beta peptide level reaching a plateau relatively early in both PET and CSF in the continuum of AD, it was not considered a good marker of progression, but it is an excellent biomarker to evaluate the response to new anti-amyloid drugs [42]. Other proteins, such as GFAP, particularly in plasma [39], or elevated CSF levels of proinflammatory cytokines MCP-1 or chitinase-3-like protein 1 (YKL-40), are currently under evaluation as markers of disease progression [43]. In the current context dominated by recent anti-amyloid therapies, it seems likely that an amyloid beta signature biomarker will become mandatory and that combination with other biomarkers of neurodegeneration or neuroinflammation will become standard diagnostic practice.

## 3. Pathological Processes Involved in Alzheimer’s Disease as a Source of Current and New Biomarkers

Alois Alzheimer, in 1907, reported, for the first time, the two pathological characteristics of AD, neuritic plaques and NFTs, analyzed from the autopsy of a 55-year-old woman who suffered from a progressive cognitive and behavioral disorder. Since then, the study of these hallmarks has advanced over the years [44]. However, the definitive diagnosis of AD requires histopathological confirmation obtained by biopsy or autopsy [15,31]; neuritic plaques and NFTs are not the only cause of the AD pathological process. The cause of global neurological deterioration is the sum of multiple processes that simultaneously coexist. In this section, we will identify several of these processes, and we will describe the search for new biomarkers related to them.

### 3.1. Neuritic Plaques

Neuritic plaques, also known as senile plaques, are deposits of Aβ that accumulate in the extracellular space around neuronal and glial cells. Amyloid beta is a 4.2 kD peptide that is obtained from the cleavage of the amyloid precursor protein (APP) [45,46]. Aβ is transmembrane generated or in endosomal compartments [47]. The 40-amino acid amyloid beta (Aβ40) peptide is also generated, becoming more abundant than the insoluble 42-amino acid form (Aβ42). Smaller amounts of shorter Aβ40 peptides, such as Aβ38 and Aβ37, are also secreted [46].

Amyloid beta monomers assemble and form a variety of oligomer species that aggregate to form protofibrils that subsequently elongate and mature into insoluble fibrils. These fibril aggregates become resistant to proteolytic cleavage [48]. At physiologic concentrations, Aβ monomers appear to be nontoxic, and oligomers rather than fibrils are the main neurotoxic agents. The properties and toxic effects of amyloid beta are varied. It destabilizes the membrane by generating pores or channels that cause an imbalance of ions, such as calcium. Aβ also induces functional and morphological alterations in synapses and alters synaptic plasticity. Moreover, amyloid beta causes mitochondrial damage, increases oxidative stress, and promotes apoptosis [48].

### 3.2. Neurofibrillary Tangles

NFTs are composed of paired helical filaments (PHFs) of highly phosphorylated forms of tau protein, a microtubule-associated protein of 68 kDa that acts in the assembly and stabilization of microtubules. Tangle formation begins with increased phosphorylation of tau, which assembles into pre-tangles and accumulates in the somatodendritic domain. Subsequently, PHFs are assembled and aggregated aberrantly, resulting in the intracellular formation of NFTs and causing disturbances in the microtubule network. Consequently, neurons degenerate and die, releasing NFTs into the extracellular space [49,50]. Deposition of insoluble tau in NFTs depends on dysregulated tau phosphorylation. Normally, there are two phosphates per tau molecule; however, in the brains of patients with AD, the presence of eight phosphates per tau molecule has been observed. Altered phosphorylation causes changes in the conformational state of tau, its microtubule detachment, and its propensity for aggregation [51]. Furthermore, exposure to tau oligomers is associated with alterations in synaptic plasticity and memory [52]. In addition, as previously mentioned, tau (t-tau and p-tau) in CSF is increased in patients with AD [14]. Total tau CSF levels are about three times higher in AD patients than in controls [32]. Moreover, there is a correlation between CSF p-tau concentrations in CSF, scores of NFTs, and hyperphosphorylated-tau load in all neocortical regions [34]. Interestingly, the combination of results on p-tau and Aβ42 obtained from CSF increased the detection and precision for detecting AD. The p-tau/Aβ42 ratio has a sensitivity of 91.6% and a specificity of 85.7% for AD diagnosis [32].

### 3.3. Neuroinflammation in Alzheimer’s Disease

Dysregulation of the immune system is a cardinal feature of AD. The possibility that immune dysfunction is involved in AD was first recognized in Alois Alzheimer’s original 1907 article, in which he described morphological alterations in glia in autopsy samples from AD brain patients [53]. Neuroinflammation is not a passive system activated by emerging senile plaques and NFTs but instead contributes to pathogenesis as much as plaques and tangles do by themselves. Colocalization of immune cells with plaques is a well-recognized neuropathological feature of AD. From a clinical perspective, an inverse relationship has been reported between cognitive performance and microglial activation but not with brain amyloid load in individuals with AD (Table 2) [21]. Imaging studies evaluating brain connectivity in people with AD implied an association between microglial activation and disruption of intrinsic functional connectivity as well as white matter microstructural breakdown [54]. Meta-analyses of observational and epidemiological studies have indicated that dysregulation of inflammatory markers is present in AD (Table 2) [20] and that this dysregulation was associated with an increased risk of developing all-cause dementia. Other systematic reviews reported data from 175 patients that have been meta-analyzed. Elevated peripheral interleukin (IL)-1β, IL-2, IL-6, IL-18, interferon-γ, homocysteine, high-sensitivity C reactive protein, C-X-C motif chemokine-10, epidermal growth factor, vascular cell adhesion molecule-1, tumor necrosis factor (TNF)-α converting enzyme, soluble TNF receptors 1 and 2, α1-antichymotrypsin and decreased IL-1 receptor antagonist, and leptin were found in patients with AD compared with healthy controls. IL-6 levels were inversely correlated with mean Mini-Mental State Examination (MMSE) scores (Table 2) [22].

Regarding cellular players, the innate immune cells involved in this process are primarily microglia and astrocytes, but other cell types, like infiltrating blood cells, also contribute to neuroinflammation, especially when the blood–brain barrier (BBB) is affected by biochemical or mechanical damage. Astrocytes work closely with microglia and could mediate some of the toxic effects of microglia in disease states. Astrocytes are specialized glial cells that form the scaffold of the entire CNS. The processes of astrocytes contribute to the BBB along with endothelial cells and closely enveloped synapses. The main functions of astrocytes include the regulation of cerebral blood flow, the maintenance of fluid and neurotransmitter homeostasis, the induction of synapse formation, and the metabolic and neurotrophic support for synapses. In AD, astrocytes release various pro-inflammatory molecules after exposure to Aβ (i.e., cytokines, interleukins (ILs), complement components, nitric oxide, and other cytotoxic compounds) and thus, ultimately, amplify the neuroinflammatory response. Human neuropathological studies conducted on AD brains report the presence of cytoplasmic inclusions of non-fibrillar Aβ in astrocytes, reflecting a phagocytic engulfment of extracellular Aβ deposits [55]. Moreover, this phagocytic activity is mediated to some extent by the ApoE genotype, so while ApoE ε2, a protective allele against AD, enhances their phagocytic capacity, ApoE ε4 excerpts a reduction in astrocytes’ phagocytic capacity, leading to faster accumulation of senescent synapses [56].

Microglia is the other main cell type involved in AD neuroinflammation. Two main phenotypical categories of microglia cells are present in the brain: resting (or quiescent) and activated. Activated microglial cells are typical pathophysiological features of AD and other neurodegenerative diseases [57]. Under physiological conditions, microglia may contribute to Aβ clearance as well as limiting plaque growth and accumulation. Microglia can bind to soluble Aβ oligomers, protofibrils, and insoluble Aβ fibrils through many cell surface receptors, including the class A1 scavenger receptor (SCARA1), cell surface cluster of differentiation (CD) markers (CD36, CD14, CD47), α6β1 integrin, and Toll-like receptors [58]. However, Aβ species may induce chronic neuroinflammation by stimulating the microglia to release pro-inflammatory cytokines and interfering with the synthesis of anti-inflammatory cytokines. Tumor growth factor β1 (TGF-β1) and tumor necrosis factor-alpha (TNF-α) are cytokines that exert opposite effects on microglia: while TGF-β1 has an anti-inflammatory effect [59], under pathological conditions like AD, TNF-α is chronically released by activated microglia, neurons, and astrocytes, and, in a vicious cycle, increased levels of extracellular Aβ stimulate its release [60]. TNF-α can stimulate γ-secretase activity, resulting in increased synthesis of Aβ peptides and a further increase in TNF-α release [61].

The important role of neuroinflammation in AD is also supported by findings that genes for immune receptors, including TREM2 and CD33, have been associated with AD in massive GWAS studies [62]. The microglial receptors CD33 and TREM2 modulate microglial pathology and neuroinflammation. TREM2 variants like p. R47H reduce the Aβ phagocytic ability of microglia and increase the risk of AD by two to four times, disrupting the microglial function and reducing the progression of the disease [63]. On the other hand, CD33 opposes the effects of TREM2 signaling and acts as the microglial phagocytosis negative regulator downstream of TREM2 [64]. Both have emerged as targets for drug development in AD.

In addition to its key role in the activation of the inflammatory response through its own phagocytic capacity and the release of cytokines in the presence of amyloid beta, the recruitment of the pyrin domain-containing 3 (NLRP3) inflammasome has also been revealed to be a crucial event in the pro-inflammatory response of microglia that is followed by the expression of caspase 1 and maturation of IL-1β1. During NLRP3 inflammasome activation, the receptor protein NLRP3 forms a multi-protein complex with apoptosis-associated speck-like protein (ASC) and caspase-1 that finally leads to neuronal death. Aβ and p-tau pathology are known to drive NLRP3 inflammasome activation, suggesting a vicious cycle of increasing pathology and neuroinflammation [65]. It has been reported that microglia, astrocytes, and neurons express components of the NLRP3 inflammasome, like caspase 4, caspase 8, and cleaved gasdermin D [66].

In summary, there is growing evidence that inflammatory processes are fundamental in the pathophysiology of AD. It is possible that in the initial stages of the disease, they manage to contain the progression of the pathological process locally, but once they are overwhelmed, they probably contribute to a vicious circle in which there is a major inflammatory response, more synthesis of amyloid beta species and major progression of the damage caused by hyperphosphorylated tau.

### 3.4. Impaired Neurogenesis in Alzheimer’s Disease

Generation of neurons *de novo* by adult neural stem cells, neurogenesis, is a process that, until the middle of the last century, was thought not to occur during adulthood. However, it is now known that adult neurogenesis occurs primarily in two areas of the mammalian CNS: in the anterior part of the subventricular zone (SVZ), along the lateral ventricles, and in the subgranular zone (SGZ) of the dentate gyrus in the hippocampus [67,68,69]. Although the production of new neurons occurs throughout life, it decreases with aging [70]. In AD transgenic animal models, particularly in 3xTg mice, neurogenesis in the hippocampus is markedly altered. The reduction in neurogenesis in the SVZ and the dentate gyrus is directly associated with the presence of amyloid beta, which is exacerbated in aged mice (Table 2) [24,27]. In addition, impaired neurogenesis precedes plaque and tangle formation, suggesting that deficits in adult neurogenesis may mediate premature cognitive decline in AD [71]. In another model of transgenic mice harboring a familial Alzheimer’s disease linked mutant APPswe/PS1DE9, a significant reduction in the proliferation and differentiation of neural progenitor cells has also been documented. These alterations are related to high levels of hyperphosphorylated tau protein in neurogenic niches (Table 2) [25]. 

The study of tissues obtained from autopsies revealed that hippocampal neuronal precursors isolated and subsequently cultured had lower viability and reached senescence earlier when precursors were derived from patients with AD compared with age-matched control subjects [72]. Likewise, another study using two different markers of neuronal progenitors reported a significant decrease in neuronal precursors in the SVZ of AD patients labeled with Mushashi but an increase when Nestin was used [73]. In a more recent study using doublecortin as a marker of immature neurons, it was reported that maturation and the neuronal number in the dentate gyrus decrease progressively in AD patients compared with neurologically healthy individuals (Table 2) [23]. Consistent with these results, a reduction in neuroblasts was also observed in mild cognitive impairment, and it was suggested that a greater number of neuroblasts is associated with better cognitive status [74]. However, there is also contradictory evidence that reports an increase in protein markers of immature neurons in the hippocampus of individuals with AD, suggesting enhanced neurogenesis [75]. These findings suggest that the increase in the production of new cells might represent a compensatory mechanism to replace the neuronal loss caused by the degenerative process. Subsequently, as the disease progresses, the regenerative capacity decreases.

### 3.5. Alzheimer’s Disease-Related Axonal Injury and Neurodegeneration

A major challenge in the fight against AD has been the unambiguous identification of non-invasive blood neurodegeneration biomarkers with predictive value prior to the progressive cognitive decline observed in AD patients. Neurofilament light chain (NfL) is a 61 kDa subunit constituent of a heteropolymer in neurofilament (Nf), a CNS cell-type specific protein, and a dominant component of the axonal cytoskeleton that have been thoroughly studied as a surrogate axonal degeneration marker in neurological diseases [76]. In a comprehensive Nature Medicine News & Views paper dated four years ago, Zetterberg and Schott pointed toward one crucial study identifying NfL as an AD blood biomarker related to axonal injury and neuronal degeneration [77]. In this study, NfL blood serum concentrations were measured by using a single-molecule array immunoassay protocol in patients carrying AD autosomal-dominant mutations in genes encoding APP, PSEN1, and/or PSEN2 proteins and compared to control individuals, showing elevated levels around 6 years preceding AD symptoms onset (Table 2) [19]. Remarkably, data from a follow-up longitudinal analysis with a mean observation time of 3 years indicated that serum NfL dynamics were coupled to brain changes and disease progression in this cohort of AD-mutation carriers [19]. Further research has confirmed these data. One study has tested blood serum NfL concentrations in MCI and sporadic AD patients enrolled in an AD Neuroimaging Initiative (ADNI) study, confirming how high NfL plasma levels were related to cognitive defaults, low cortical glucose metabolism, brain hippocampal atrophy and ventriculomegaly [78]. In turn, other work has compared serum NfL levels between MCI and cognitively unimpaired participants, showing that a higher NfL concentration represents a major risk in the transition from a normal to a cognitive default stage [79]. Finally, a very recent work evaluating connectome-wide associations in autosomal-dominant AD mutation carriers versus controls has demonstrated a positive correlation between NfL and a deterioration of the functional connectivity within the default mode network, a set of brain regions that have been associated with cognitive function [80]. All these scientific outcomes give evidence of NfL as a convenient blood marker to evaluate cognitive decline, disease progression, and alterations in brain functional connectivity in an AD context.

### 3.6. Hippocampal Degeneration in Alzheimer’s Disease

In AD, besides the abnormal deposition of amyloid beta and tau in the neocortex, atrophy of the hippocampus is a pathological feature of the disease [81]. The hippocampus is critical in episodic memory and is a site of new neuron formation in the adult brain [82]. Specifically, neuronal stem cells located in the dentate gyrus SGZ give rise to differentiated neurons in the hippocampus by a mechanism dependent on SHH ciliary transduction [83]. In mice, conditional loss of the intraflagellar transport 88 protein (Ift88), a critical regulator of cilia formation in the cortex and hippocampus, results in deficits of aversive and recognition memory. These animals also display an altered paired-pulse response in the CA1 hippocampus, suggesting the participation of neuronal cilia in synaptic activity [84]. Interestingly, a recent discovery unveils that a group of neurons within the brainstem and hippocampus establish a cilia–axonal synapse [85]. The work describes that serotonergic axons from the raphe nuclei contact and activate ciliary 5-HT6 in CA1 pyramidal neurons, modifying chromatin accessibility. These findings shed light on the potential role of primary cilia (PC) in neuromodulation. The analysis of AD mouse models has revealed an intriguing link between PC dynamics and cognitive function. The 3xTg-AD mouse model that carries mutations in APP, Tau, and PSEN1 exhibits impaired neurogenesis in the hippocampus SGZ, and dentate granular cells display a reduction in cilia length. Notably, this impairment in neuronal proliferation is exacerbated in females and aging mice (Table 2) [27]. In the APP/PS1 model, there is an abnormal elongation of 5-HT6-positive cilia on hippocampal neurons, along with an increase in the length of the axonal initial segment. Furthermore, behavioral analysis reveals that the administration of a 5-HT6 antagonist leads to a significant amelioration of the cognitive impairment observed in these mice (Table 2) [28]. In vitro models show that neuronal cilia length reduces upon treatment with the Aβ42 fragment [86]. Similar phenotypes are reported in cilia of NIH3T3 fibroblasts treated with Aβ42, along with a decrease in the transmission of SHH signaling response [87].

In summary, these observations underscore the emergent significance of PC as an organelle involved in the maintenance and functionality of the hippocampus. Nevertheless, additional research endeavors are required to provide mechanistic insights into how ciliary dysfunction contributes to the pathology of AD.

### 3.7. Synaptic Dysfunction in Alzheimer’s Disease

Synaptic dysfunction and synapse loss are considered early events in AD pathophysiology [88,89]. For this reason, synaptic damage biomarkers have emerged as suitable candidates for early AD diagnosis [90,91]. In this context, neurogranin (Ng) is a 7.5 Kda small post-synaptic protein located in dendrites and dendritic spines in hippocampal and cerebral cortex neurons [92,93], playing a crucial role in synaptic plasticity, postsynaptic sensibility, and LTP induction [94,95]. All these characteristics pointed to Ng as a good measurable indicator of synaptic dysfunction. Ng levels can be detected and quantified in CSF [96] and these levels are elevated in AD and MCI patients in comparison to cognitively unimpaired aged controls (Table 2) [17]. Interestingly, CSF Ng level increases are correlated with increases in other synaptic proteins such as PSD-95 and SNAP-25 in AD patients compared with other non-AD neurodegenerative disease patients [91] and have also been significantly correlated with t-tau and p-tau levels as well as with the MMSE state examination in AD patients [97]. A recent paper working with aged postmortem human brain tissues has revealed significantly reduced Ng levels in AD patients with respect to healthy individuals across all brain regions (Table 2) [18]. These findings are relevant as they strongly suggest that the increased level of Ng previously observed in the CSF of AD patients might reflect the loss of Ng from the patient’s pathological brain. Remarkably, the authors’ findings have also revealed that the amount of synapse-associated Ng is associated with cognitive decline [18], validating previous data from other groups [98]. All these works confirm CSF Ng level measurements as an early synaptic default biomarker of AD and a predictor of cognitive decline.

Concerning Ng levels detection in blood, a work from De Vos and colleagues targeting the C-terminal part of Ng was pivotal to detecting Ng levels in plasma, albeit they did not observe serum Ng differences between AD patients and control samples [99]. More recently, another group has adopted a different strategy by measuring Ng concentration in blood exosomes—extracellular vesicles secreted by cells into the circulating blood—from MCI and AD patients, observing a decreased level of Ng in blood exosomes in both MCI and AD patients compared to control individuals (Table 2) [16]. In addition, the Ng amount was highly associated with the level of cognitive decline of the subjects under study [16]. In spite of this promising result and the potentiality of this approach [100], more future work is needed prior to considering blood Ng level changes as a useful AD biomarker.

### 3.8. Alzheimer’s Disease-Dependent Neuronal Excitability Dysregulation

Calmodulin (CaM) is the major calcium-binding protein that regulates the activity of CaM-binding proteins (CaMBPs). This phylogenetically conserved protein plays a major multifunctional role in neuronal calcium signaling. CaM is a central integrator of synaptic plasticity with properties that regulate signal transduction required for long-term potentiation (LTP) and long-term depression (LTD), which are usage-dependent changes in synaptic efficacy [101]. Depending on the intensity and duration of the intracellular calcium flux, CaM can opt for bidirectional and opposite functions. Through a large but short Ca+ signal, CaM can activate CaMKII, which, in turn, phosphorylates ARMPA-R, which correlates with the induction of LTP. On the other hand, with a Ca+ flux of long duration but of moderate amplitude, CaM can activate a phosphatase-producing AMPA-R depression through dephosphorylation and consequent LTD induction [102].

CaM immunoactivity is decreased or lost in AD postmortem human brains (Table 2) [26]. The calmodulin hypothesis in AD was derived from the calcium hypothesis, which proposes that dysregulation of intracellular Ca+ homeostasis and neuronal excitability are both altered as common pathways in the pathology underlying AD [103,104]. Furthermore, CaM is related to several of the processes associated with the pathophysiology of AD. Many of the proteins involved in the amyloidogenic pathway in AD are CaMBPs, including amyloid beta protein precursor, β-secretase, presenilin-1, and ADAM10 [105]. CaM appears to have a role in the regulation of APP, promoting its catabolism through the non-amyloidogenic pathway [106]. Regarding the tau protein processing, CaM interacts with a tubulin binding site placed on tau [107], and many CaMBPs are involved in the processing of NFTs, including the phosphatase calcineurin, and the kinases CaMKII and CDK5 [105]. Calcineurin also binds to tau, and CaM interferes with this binding, affecting the dephosphorylation of this protein by calcineurin contributing in this form to the tau phosphorylation/dephosphorylation balance [108]. Importantly, CaM can act as a buffer for intracellular Aβ peptides, decreasing the formation of Aβ fibrils and plaques [109]. Therefore, it is expected that the decrease in regulation of CaM makes neurons more prone to suffer toxicity induced by free Aβ peptides.

### 3.9. Acetylcholine Dysregulation in Alzheimer’s Disease

The cholinergic system is strongly altered in AD. Acetylcholine, the first neurotransmitter to be discovered, is involved in multiple physiological processes such as learning, memory, and attention [110,111,112]. In AD, cholinergic neurons degenerate in the nucleus basalis of Meynert, contributing to the memory loss shown in these patients [113]. Furthermore, acetylcholine transcription is severely decreased in the remaining cholinergic neurons, further leading to a global decrease in cholinergic activity [114]. This prominent and well-known cholinergic alteration of AD prompted the development of inhibitors of acetylcholinesterase—enzymes that degrade acetylcholine—which, until now, are reference therapies for AD [115]. AD biomarkers based on the cholinergic system have also been evaluated. Interestingly, there are significantly lower levels of acetylcholine in the CSF of patients with AD compared to controls (Table 2) [29]. Recently, new technologies have been developed, such as the application of a thioflavin T@Er-MOF ratiometric fluorescent sensor for the highly sensitive detection of not only acetylcholine in CSF but also presenilin 1 and amyloid beta [116]. In addition, the usefulness of CSF acetylcholinesterase as a biomarker that evaluates pharmacological effects in AD has been highlighted. After prolonged treatment with acetylcholinesterase inhibitors (donepezil and galantamine), a marked increase in CSF acetylcholinesterase occurs. This increase is dose-dependent, being more relevant in patients with a better clinical effect of donepezil [117]. With respect to acetylcholine receptors, postmortem studies have shown that elevated binding of the astrocytic α7 subunit of nicotinic acetylcholine receptors (α7nAChR) is associated with greater amyloid beta plaque pathology in the superior frontal cortex, regardless of the clinical status of AD patients [118]. Therefore, α7nAChR has been proposed as a new early biomarker with different PET imaging tracers [119].

## 4. Genes Involved in Alzheimer’s Disease Pathophysiology

### 4.1. The E4 Allele of Apolipoprotein E (APOE4) as a Major Alzheimer’s Disease Genetic Risk

ApoE is a polymorphic glycoprotein of 299 amino acids encoded by the APOE gene on the long arm of chromosome 19 in humans. A single gene locus has three major alleles: ε2, ε3, and ε4. Thus, there are three isoforms of this protein, ApoE2, ApoE3, and ApoE4, that differ from one another only by single amino acid substitutions [120]. ApoE has varied functions involved in regulating lipid transport from one tissue to another, glucose metabolism, neuronal signaling, neuroinflammation, and mitochondrial function. In the CNS, ApoE is mainly produced by astrocytes and transports cholesterol to neurons via ApoE receptors, which are members of the low-density lipoprotein receptor (LDLR) family [121]. The APOE ε4 allele is the major genetic risk factor for AD for late-onset, sporadic AD, and decreases the age of onset by 10 to 15 years [122,123]. In contrast, the APOE ε2 allele appears to be a protector factor concentration [123,124]. ApoE isoforms influence the clearance and deposition of Aβ and depend on the isoform of ApoE expressed [35,123]. 

### 4.2. Role of Genes for β-Secretases (BACE1 and BACE2), Amyloid Precursor Protein (APP), Presenilin-1 (PSEN1) and Presenilin-2 (PSEN2) in Alzheimer’s Disease

Deciphering the specific role of the elements involved in Aβ generation is pivotal to understanding amyloid aggregation in AD. As a first step in the process, APP is generated from the endoplasmic reticulum (ER) and transported across the Golgi apparatus to the plasma membrane [125]. The transmembrane APP (100–140 kDa) can be cleaved either by α-secretases, like ADAM10 [126] in a non-amyloidogenic pathway to form a non-pathogenic soluble APP protein α (sAPPα), or by β-secretases [127] in an amyloidogenic pathway to produce a soluble APP protein β (sAPPβ) and a C-terminal fragment (CTF_99_). This CTF_99_ is later processed by γ-secretases, leading to the formation of several Aβ peptide fragments that range from small 1–13 or 1–14 peptides to 1–42, although 1–40 and 1–42 fragments are those showing a higher tendency to oligomerize, aggregate and form plaques [128]. Since the α-secretase activity of ADAM10 gives rise to the non-amyloid pathway, it has been proposed that increased activity protects against amyloid beta deposition. In vitro studies have shown that exogenous treatment with the ADAM10 variant inhibits the formation and aggregation of neuronal extracellular amyloid β [129]. Furthermore, ADAM10 is a synaptic protein involved in the structural plasticity of dendritic spines. Therefore, the positive regulation of ADAM10 activity as a therapeutic strategy for Alzheimer’s disease has great potential that needs to be further evaluated [130]. 

As mentioned previously, Aβ42 levels in CSF and plasma have been considered as classical biomarkers, together with p-tau, for potential diagnosis, therapy follow-up, and prognosis in AD patients. However, other players of the amyloidogenic pathway have also been studied as alternative biomarkers of this neurodegenerative disease. Beta-secretase 1 (BACE1) represents the rate-limiting catalytic step for Aβ production [127]. Inhibitors of BACE 1 and of its close homolog, BACE2, have been thoroughly tested, aiming to interrupt and/or delay amyloid beta pathology in AD. Unfortunately, their potential therapeutic use in a clinical setup for AD patients has been notably hampered by undesired mechanism-based side effects [131]. Interestingly, BACE1 activity rates and protein concentration levels quantified in CSF and plasma fluids are all increased in the brains of AD and MCI patients [132,133,134]. Moreover, a recent report has correlated increases in BACE1 activity with levels of Aβ40, Aβ42, and Aβ40/42 ratio in serum [135]. This work points toward considering plasma BACE1 level changes, together with the above-mentioned Aβ markers, as complementary biomarkers in AD clinical trials [135].

The γ-secretase enzyme is a transmembrane complex formed by two constant subunits—nicastrin and presenilin enhancer 2 (PEN-2)—and two variable subunits: anterior pharynx-defective 1 (APH-1, APH1a, or APH1b) and presenilin (presenilin-1, PSEN1 or presenilin-2, PSEN2) as the catalytic site of the enzyme [136]. Mutations in APP, PSEN1, and PSEN2 genes cause an early onset familial form of AD (EOAD), with an autosomal dominant inheritance, accounting for 5% of all AD cases [137,138]. Multiple research efforts have been focused on the identification and characterization of pathogenic mutations on APP, PSEN1, and PSEN2 genes as causatives for (EOAD) [137]. Many studies indicate that although an APP point mutation located next to the β-secretase cleavage sites is protective, reducing age-related brain Aβ accumulation [139], others are harmful, causing EOAD [134,140,141]. In turn, PSEN1 mutations are among the major agents responsible for EOAD, with patients carrying PSEN1 mutations showing a boost in AD disease progression [142]. 

In contrast with BACE1, changes in APP, PSEN1, and PSEN2 protein levels have been considered unsuitable biomarkers in human fluids. Only two studies have shown that the levels of soluble PSEN1 complexes are increased in ventricular postmortem CSF from AD patients compared to control individuals [143] as well as in lumbar CSF samples obtained from genetically determined AD patients [144]. Regarding PSEN2, a transcript named PSV2 encoding a truncated form of PSEN2 is upregulated in AD postmortem brains [145].

In addition, the involvement of PSEN1 in multiple pathways regulating calcium homeostasis, Notch signaling, or beta-catenin stability, among others, causes PSEN1 mutations to impact other neurodegenerative but also non-neurodegenerative diseases [142,146], a fact that severely interfere with the design and development of new therapeutic avenues for AD based on PSEN1 level changes. As an example, in recent clinical trials for autosomal dominant AD, human carriers of PSEN1 mutations have been longitudinally assessed for Aβ42 and p-tau in CSF but not for APP or PSEN1/2 levels, together with clinical symptoms and imaging analysis as endpoints for drug efficacy evaluation [147].

### 4.3. Other Genes Associated with Alzheimer’s Disease Risk

New advances in the analysis of millions of genomic polymorphisms in thousands of subjects have revealed new genes associated with the risk of AD that have diverse functions in the pathology of the disease. Sortilin-related receptor L (SORL1) participates in vesicular trafficking and directs APP to endocytic pathways for recycling. Inherited variants of SORL1 are associated with late-onset AD, and loss of SORL1 is specific to the most common sporadic form of AD [148]. When SORL1 is underexpressed, APP is sorted into amyloid beta-generating compartments [149]. Depletion of the SORL1 gene in neuronal cell lines shows early endosome enlargement, an effect similar to what occurs in APP and PSEN mutation [150]. In murine models for AD, SORL1 deficiency increases amyloid beta levels and exacerbates early amyloid pathology in the brain [151]. Other genes with important roles in endocytosis have been identified in various GWAS studies, including BIN1, PICALM, CD2AP, and EPHA1 [152,153].

Furthermore, several loci in ABCA7, encoding the ATP-binding cassette (ABC) transporter A7, have also been recognized as novel risk factors for AD [154]. ABCA7 promotes cellular cholesterol efflux to apoE discs and regulates APP processing, resulting in an inhibition of amyloid beta production [155]. The AD susceptibility loci rs3764650 of the ABCA7 gene is associated with neuritic plaque burden [156], and the minor allele is associated with age of onset and disease duration in late-onset AD [157]. It has also been shown that in the mouse model of AD, the deletion of ABCA7 doubles the accumulation of brain amyloid beta [158]. Similarly, in relation to cholesterol metabolism, single nucleotide polymorphisms of CLU, encoding the apolipoprotein clusterin that confer protection against late-onset AD, have been identified [152,153].

## 5. Olfactory Neuroepithelium as a Cellular Model to Evaluate and Identify Biomarkers

### 5.1. Olfactory Dysfunction in Alzheimer’s Disease

Alterations in olfactory functions are suggested to be an early characteristic in neurodegenerative diseases. Olfactory dysfunction is quite frequent in AD, being present in 85% of patients with early stages of this disease [159]. Olfactory identification is more affected in patients with AD than in those with MCI [160]. The precise mechanism of this dysfunction in AD is not fully elucidated. It is suggested that this alteration in the perception of smell may be due in part to the accumulation of the pathological marker’s characteristic of AD in cerebral olfactory circuits (Figure 1) [161,162,163,164,165,166,167]. In autopsy studies, impaired odor identification has been associated with plaques and tangles present in the entorhinal cortex, the olfactory bulb, and the CA1/subiculum area of the hippocampus [168]. Olfactory impairment predicts the incidence of MCI and progression from MCI to AD [169,170]. A below-average on the odor identification score increases the risk of developing MCI by 50% at five years [171]. Different studies suggest that a simple odor test is valuable for screening individuals with MCI and MCI likely to progress to AD [160,170].

### 5.2. Characteristics of the Olfactory Neuroepithelium

To evaluate cellular processes and molecular changes that occur in human neural cells more accurately, ONE has been assessed as a study model. The ONE is a pseudostratified columnar epithelium located in the posterior region of the nasal cavity, along the ventral surface of the cribriform plate, the superior turbinate, and extending to the middle turbinate. It consists of four main cell types: supporting cells (microvillar and sustentacular cells), basal cells, and bipolar olfactory sensory neurons (OSNs). Hence, it is the only source of neuronal cells that can be obtained from a living human being [172,173]. Furthermore, the ONE is one of the few regions where neuronal regeneration exists continuously throughout human adult life. The cell population present in the ONE is in different stages of maturation, from steam cells and progenitor cells to immature and mature neurons [174]. Basal cells divide asymmetrically, giving rise to neuronal progenitors that subsequently differentiate into mature OSNs, which have a half-life of 4 to 6 weeks [175]. The cells of this tissue can differentiate not only into neuronal lineage populations but also into glial lineage depending on environmental signaling in vitro [176]. The precursors derived from ONE express receptors for neurotrophins (Trka receptors) on their surface, which is consistent with neural properties [176]. Stimulation with neurotrophic factors results in the loss of progenitor characteristics and the gradual gain of properties of mature neurons and neurite formation [177]. Another important characteristic present in the precursors of the ONE is the expression of neurotransmitter receptors, including receptors for dopamine, serotonin, and glutamate, and the signaling mechanisms of these neurotransmitters are active [178]. ONE cells are obtained either from postmortem tissues, biopsies, or through nasal exfoliation [179]. Nasal exfoliation consists of introducing a small brush into the nasal cavity at the level of the middle turbinate, and with circular movements, a heterogeneous cell population is obtained. This technique facilitates obtaining ONPs from ONE compared to biopsies since it is a non-invasive outpatient procedure that does not require anesthesia (Figure 2) [180].

### 5.3. What Biomarkers Have Been Studied in the Olfactory Neuroepithelium (ONE) for Alzheimer’s Disease?

In the search for Alzheimer’s biomarkers in ONE, research began by studying postmortem samples. In 2009, Arnold et al. (Figure 3) obtained the ONE of the osseous septa and the cribriform plate from subjects with AD confirmed by autopsy. They found that AD patients had higher amyloid-β expression in this tissue, mostly located in the superficial regions. Similarly, the expression of paired PHFs was over-expressed. These pathological findings of amyloid-β and PHFs in the ONE correlated with the average amyloid plaques and neurofibrillary PHFs found in cortical regions. Amyloid-β was evident in 71% of AD cases compared to 22% of normal cases and 14% of cases with other diseases. PHF-tau was present in 55% of cases with AD, 34% with normal brains, and 39% with other neurodegenerative diseases [181]. Contracting results were obtained by Godoy et al., (Figure 3) where they did not find increased amyloid-β and PHF expression in the ONE and did not correlate with brain pathology scores at autopsy [182]. These two studies have contradictory findings, probably due to the cellular and molecular changes that can occur due to the time between death and sample collection. Therefore, it highlights the need to obtain antemortem samples. In this regard, in a study where ONE was obtained by biopsy through nasal endoscopy (Figure 3), the authors studied ONE samples of three groups: patients without cognitive impairment (NCI), with MCI, and with AD. They reported that the amyloid-Aβ deposits were more expressed in cells near the ONE lumen of patients with MCI and AD. Additionally, they reported that in those samples derived from AD and MCI, the cell migration is diminished compared to those from NCI [183].

Employing a different methodology, Riquelme et al. (Figure 3) obtained ONPs from ONE by nasal exfoliation of living individuals. They found an increase in levels of t-tau and p-tau in ONP precursor cultures from AD patients in comparison with control patients without AD. In addition, the t-tau was distributed in a punctate pattern in an increased number of olfactory neuronal precursors of AD [184]. Nasal exfoliation consists of introducing a small brush into the nasal cavity at the level of the middle turbinate, and with circular movements, a heterogeneous cell population is obtained. Subsequently, the sample collected is cultured in a selective medium that restricts the growth of non-neural cells, boosting the development of ONPs. This powerful technique facilitates obtaining ONE compared to biopsies. It represents a non-invasive outpatient procedure that does not require anesthesia and allows the massive amplification of the neuronal lineage population as a model to evaluate cellular mechanisms underlying AD pathology. In addition, it allows monitoring of the disease through longitudinal studies of the stages of AD progression, from MCI to the late stage.

## 6. Discussion

Although the main biomarkers studied in AD are protein tau and amyloid-beta peptide, it is important to understand that this condition is multifactorial. Multiple pathological processes occur and are part of the global process of neurodegeneration. Understanding these processes leads to the search for new elements that could be useful in the study of biomarkers. In this review, some of these biomarkers were mentioned, some already widely studied and others with great potential to be evaluated more exhaustively. Furthermore, in this era of biomarkers, it is also essential to look for new models for their study. The biomarkers accepted in the AD diagnostic criteria are not very accessible to the common clinical environment, so we should choose to analyze biomarkers that are affordable. Therefore, we propose neuronal precursors derived from the ONE as a unique model in which neuronal lineage cells can be obtained from living patients. In these cells, it is possible to study the different biological and molecular changes that underlie neurodegenerative diseases such as AD, including some altered processes mentioned in this review.

## Figures and Tables

**Figure 1 brainsci-14-00046-f001:**
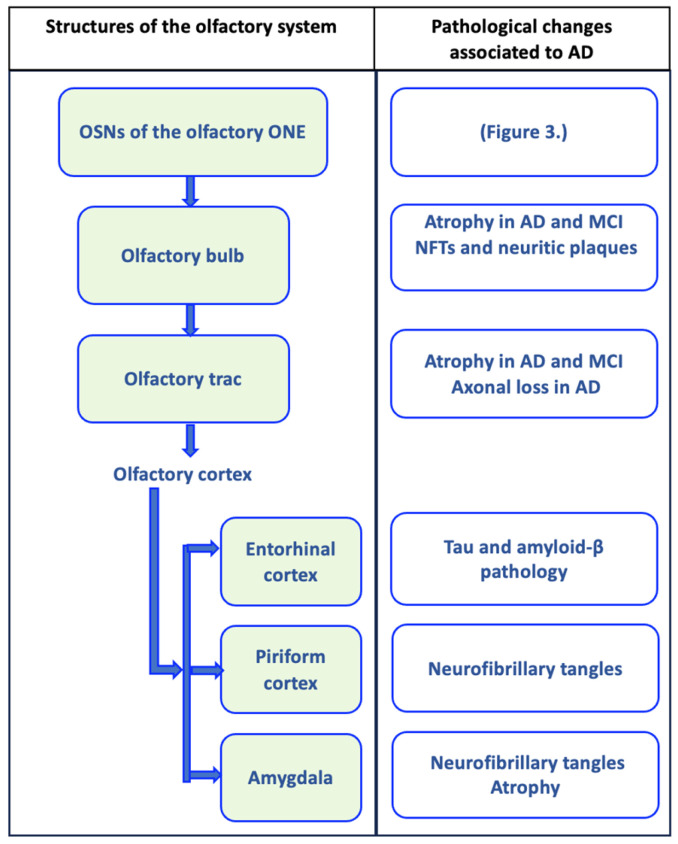
Olfactory circuit and pathological changes associated with Alzheimer’s disease (AD). The olfactory circuit begins with olfactory sensory neurons (OSNs) of the olfactory neuroepithelium (ONE), whose axonal extensions form olfactory nerve fibers that cross the cribriform plate of the ethmoid bone to penetrate the olfactory bulb, where they establish synapsis with mitral cells. Mitral cells send their axonal projections in a bundle that forms the olfactory tract, which connects the olfactory bulb with the upper olfactory areas. The olfactory tract is divided into the medial olfactory striae, whose axons go to the fornix, and the lateral olfactory striae, which connect with the olfactory cortex (piriform cortex, entorhinal cortex, and the amygdala). In each of these structures, pathological changes associated with AD have been evident, and they may even be present in mild cognitive disorder (MCI), as represented in the figure [161,162,163,164,165,166,167].

**Figure 2 brainsci-14-00046-f002:**
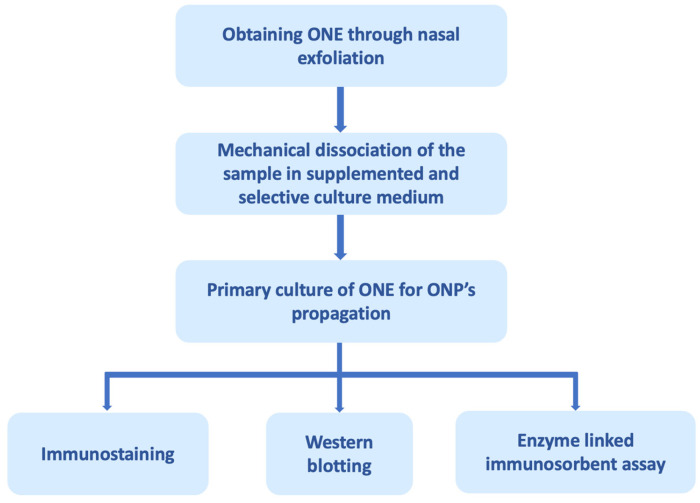
Olfactory neuroepithelium (ONE) processing through nasal exfoliation for biomarker analysis. ONE can be obtained through autopsies, biopsies, and, recently, by nasal exfoliation. The figure exemplifies obtaining it through nasal exfoliation, in which cells of the sample are mechanically dissociated in a selective culture medium that promotes the proliferation of olfactory neuronal precursors (ONPs) and restricts the growth of other cell populations present in the ONE. The primary culture of the ONPs can subsequently be processed for analysis by immunostaining, Western blotting, enzyme-linked immunosorbent assay, among others [180].

**Figure 3 brainsci-14-00046-f003:**
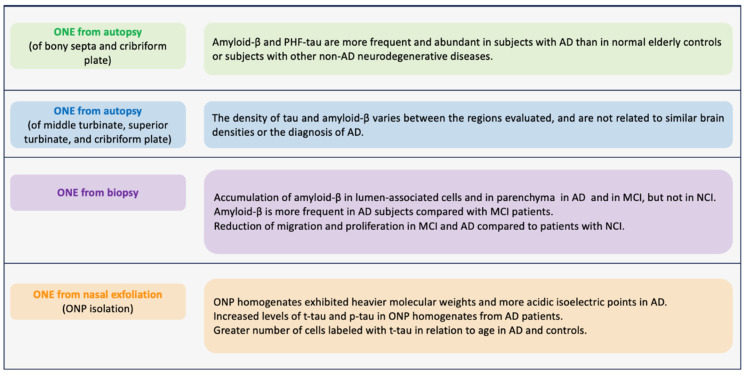
Major biomarkers observed in olfactory neuronal precursors (ONPs) derived from persons diagnosed with Alzheimer’s disease (AD). This figure summarizes the presence of AD biomarkers and pathological AD-related processes in samples of olfactory neuroepithelium (ONE) from subjects with AD. Tissues were obtained by autopsy, biopsy, or ONPs obtained by nasal exfoliation. References in order: [181,182,183,184].

**Table 1 brainsci-14-00046-t001:** Biomarkers included in diagnostic criteria and their use in clinical practice.

NIA-AA Criteria [4]		
Probable or possible AD dementia with evidence of biomarkers for the pathophysiological process of AD.	Markers of amyloid deposition.	Decrease in CSF amyloid beta levels. Positive PET amyloid imaging.
	Markers of neuronal damage.	Increased CSF tau (total tau and phosphorylated). Decreased FDG-PET consumption in the temporoparietal cortex. Atrophy in the medial, basal, and lateral temporal lobe temporoparietal cortex and in the medial parietal cortex on MRI.
* Mainly used for research purposes. * They increase diagnostic certainty but do not confirm the diagnosis.		
**DSM-5 Criteria [3]**		
Major neurocognitive disorder due to probable AD.	Evidence of a genetic mutation causing Alzheimer’s disease in family history or genetic testing: APP, PSEN1, or PSEN2.	
Positive PET amyloid imaging PET or reduction in amyloid levels in CSF may have diagnostic value. Hippocampal and temporoparietal cortex atrophy on MRI, temporoparietal hypometabolism on FDG-PET, and increased total tau and phospho-tau levels in CSF provide evidence of neuronal damage but are less specific for Alzheimer’s disease.		
* Currently, these biomarkers are not fully validated, and many are only available in tertiary care devices.		

* Use and interpretation of biomarkers in diagnostic criteria.

**Table 2 brainsci-14-00046-t002:** Main biomarkers in Alzheimer’s disease, biological processes, molecules, and organelles to be further studied as possible biomarkers.

AD Biomarkers and Processes	Human Blood	Human CSF	Other Human Tissue/Technique	Additional Data from Mouse Model
Amyloid-β		↓ [14]	↑ brain tissue *postmortem* (neuritic plaques) [15] ↑ PET [10]	
Tau protein		↑ [14]	↑ brain tissue *postmortem* (neurofibrillary tangles) [15]	
Neurogranin	↓ * [16]	↑ [17]	↓ brain tissue *postmortem* [18]	
Neurofilament	↑ [19]			
Neuroinflammation TGF-β Microglia activation		↑ [20]	↑ PET [21]	
Peripheral inflammation Leptin IL-1r agonist IL-6 IL-18 TGF-β TNF-α	↓ ↓ ↑ ↑ ↑ ↑ [22]			
Neurogenesis			↓ brain tissue *postmortem* (hippocampus) [23]	↓Proliferation and differentiation of neural progenitor cells in APP/PS1 and 3xTg-AD mouse models [24,25]
Calmodulin			↓ brain tissue *postmortem* [26]	
Primary cilia				↓ in dentate gyrus cells of 3xTg-AD mice [27] ↑ in length in hippocampus of APP/PS1 mice [28]
Acetylcholine		↓ [29]		

This Table summarizes the findings regarding the most accepted and widely described biomarkers, such as amyloid-β and tau proteins in blood and cerebrospinal fluid (CSF), postmortem human brains, and PET images. In addition, in biological processes altered in Alzheimer’s disease, molecules such as calmodulin and organelles as primary cilia are shown as sources of potential biomarkers in different human fluids and tissues, as well as in transgenic animal models of AD. ↑: increased; ↓: decreased; *: in blood exosomes.

## Data Availability

Not applicable.

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
