# Peer review of "Biomarkers in Alzheimer’s Disease: Are Olfactory Neuronal Precursors Useful for Antemortem Biomarker Research?"

_brainsci, 2024, doi:10.3390/brainsci14010046_

Round 1

Reviewer 1 Report

Comments and Suggestions for Authors

The paper is clearly written and can be read and understood easily. It contains appropriate references in this field. 

I have the following suggestions and questions:

1. I recommend the authors add appropriate references for the provided biomarkers in the Tables.

2. Were the biomarkers mentioned in the paper approved and used in clinical practice? It should be emphasized more in the text (maybe in a table).

3. It is known that AD is characterized by neuronal damage and consequently a lower level of neurotransmitter acetylcholine. Could acetylcholine level be used as a biomarker of AD? 

Author Response

Thank you very much for your inspiring observations and suggestions, they have been quite enriching for us and we have followed your recommendations:

  1. Appropriate references are now included in table and figure descriptions.

  1. In this new manuscript version we have added a new Table 1 describing all biomarkers accepted in the diagnostic criteria of the NIA-AA and DSM-5, as well as its use and interpretation in clinical practice.

  1. The alteration of in acetylcholine neurotransmission in AD is a widely studied topic. Acetylcholine levels in cerebrospinal fluid have been evaluated and they are decreased in AD. Therefore, the answer to your observation is yes. As several elements involved in cholinergic signaling could be evaluated as AD biomarkers, we have added a short section in the new version: “3.9 Acetylcholine dysregulation in Alzheimer's disease”.

Reviewer 2 Report

Comments and Suggestions for Authors

My suggestions:

1. In the abstract, authors may focus a little more on Olfactory neuronal precursors.

2. A pathway figure, which explains, how olfactory dysfunctions could impact AD onset may be helpful in this manuscript. 

3. In the genetic parts, authors may mention briefly a few more genetic factors for AD, including SORL1, ABCA7, TREM2, etc.

4.  Authors may make a workflow figure on methods of biomarker analysis in olfactory systems., 

5. Authors may briefly mention, how alpha secretases could play a role in neuroprotection or how their dysfunctions may impact neurodegeneration. 

Author Response

We appreciate all constructive criticisms from reviewer 2 that are certainly improved our manuscript.This is a point-to-point answer to your questions, observations and comments:

  1. Following reviewer’s 2 suggestions, the use of olfactory neuronal precursors as a study model for the search of biomarkers has been highlighted in the new abstract version.

  1. Indeed, olfactory dysfunctions in AD are both an effect and also part of AD pathophysiological processes. In the revised text we have now included a new figure (Figure 1) revealing the evidence of AD-associated pathology related to brain olfactory circuit structures.

  1. We have added a new section: “4.3 Other genes associated with Alzheimer’s diseases risk”. In this section we briefly describe genes involved in processes such as endocytosis (SORL1, BIN1, PICALM, CD2AP, EPHA1) and cholesterol metabolism (ABCA7, CLU) that have also been recognized as risk genes for AD. In addition, in the section “3.3 Neuroinflammation in Alzheimer's Disease” we had added text regarding genes such as TREM2 and CD33.

  1. In the new manuscript version, we have created a flow chart figure (Figure 2) highlighting the steps needed to obtain and process olfactory neuronal precursors (cells of the olfactory system accessible from living patients) using various methodologies to evaluate different biomarkers.

Round 2

Reviewer 2 Report

Comments and Suggestions for Authors

Authors fulfilled my suggestions. Thank you.